# Accuracy of a Computer-Aided Dynamic Navigation System in the Placement of Zygomatic Dental Implants: An In Vitro Study

**DOI:** 10.3390/jcm11051436

**Published:** 2022-03-05

**Authors:** Juan Ramón González Rueda, Irene García Ávila, Víctor Manuel de Paz Hermoso, Elena Riad Deglow, Álvaro Zubizarreta-Macho, Jesús Pato Mourelo, Javier Montero Martín, Sofía Hernández Montero

**Affiliations:** 1Department of Implant Surgery, Faculty of Health Sciences, Alfonso X El Sabio University, 28691 Madrid, Spain; jgonzrue@myuax.com (J.R.G.R.); igarcavi@gmail.com (I.G.Á.); elenariaddeglow@gmail.com (E.R.D.); shernmon@uax.es (S.H.M.); 2Department of Maxillofacial Surgery, Quiron Health Hospital, 28002 Madrid, Spain; vdepaz@gmail.com; 3Department of Surgery, Faculty of Medicine, University of Salamanca, 37008 Salamanca, Spain; javimont@usal.es; 4Department of Surgery, Faculty of Dentistry, University of Navarra, 31009 Pamplona, Spain; jpatomourelo@hotmail.com

**Keywords:** implantology, computer-aided surgery, image-guided surgery, zygomatic implants, navigation system

## Abstract

The objective of this in vitro study was to evaluate and compare the accuracy of zygomatic dental implant (ZI) placement carried out using a dynamic navigation system. Materials and Methods: Forty (40) ZIs were randomly distributed into one of two study groups: (A) ZI placement via a computer-aided dynamic navigation system (*n* = 20) (navigation implant (NI)); and (B) ZI placement using a conventional free-hand technique (*n* = 20) (free-hand implant (FHI)). A cone-beam computed tomography (CBCT) scan of the existing situation was performed preoperatively to plan the surgical approach for the computer-aided study group. Four zygomatic dental implants were placed in anatomically based polyurethane models (*n* = 10) manufactured by stereolithography, and a postoperative CBCT scan was performed. Subsequently, the preoperative planning and postoperative CBCT scans were added to dental implant software to analyze the coronal entry point, apical end point, and angular deviations. Results were analyzed using the Student’s *t*-test. Results: The results showed statistically significant differences in the apical end-point deviations between FHI and NI (*p* = 0.0018); however, no statistically significant differences were shown in the coronal entry point (*p * = 0.2617) or in the angular deviations (*p* = 0.3132). Furthermore, ZIs placed in the posterior region showed more deviations than the anterior region at the coronal entry point, apical end point, and angular level. Conclusions: The conventional free-hand technique enabled more accurate placement of ZIs than the computer-assisted surgical technique. In addition, placement of ZIs in the anterior region was more accurate than that in the posterior region.

## 1. Introduction

Zygomatic implants (ZIs) have proven to be a suitable treatment option in the restoration of the extremely atrophic, totally edentulous maxillae usually caused by maxillary resection in patients with oncological pathologies, congenital deformities, or metabolic disorders—patients undergoing radiotherapy—or immunosuppressed patients [1]. ZIs are especially indicated for use in patients with compromised vascularization, which can affect the outcome of bone grafts used to regenerate maxillary defects; they are also indicated in cases of incompatibility of the donor area [2]. Specifically, ZIs provide a predictable treatment option that prevents long waiting times before prosthetic rehabilitation when compared with alternative techniques for conventional implant placement using grafting materials [3]. Additionally, a recent meta-analysis reported a failure rate of 2.89% (CI-95% 1.83–3.96%) associated with conventional-length dental implants (*n* = 3549), while the failed implantation of ZIs (*n* = 1895) had an estimated incidence rate of 0.69% (CI-95% 0.21–1.16%) over a follow-up period ranging from 3 to 163 months [4].

In 1988, Branemark first described using an intrasinusal placement approach for ZIs; however, this technique can lead to sinusitis, dental implant failure, oroantral fistula, periorbital and conjunctival hematoma or edema, paresthesia, difficulty speaking, pain, and edema [5]. Alternative placement approaches that depend on bone availability have subsequently emerged, such as the extrasinusal, extramaxillary, or slot techniques [6]; however, these are not devoid of intraoperative complications, which are primarily linked to operator experience. Therefore, preoperative planning techniques using cone-beam computed tomography (CBCT) scans [7] have been recommended to enable accurate computer-aided surgery with both static and dynamic navigation systems [8]. These increase the accuracy of dental implant placement, thereby reducing the risk of intraoperative complications and maintaining high survival rates of patients receiving dental implants [9]. Computer-aided surgery using static navigation systems has been widely used for the placement of conventional-length dental implants, showing a high success rate; furthermore, static navigation techniques have shown a mean horizontal deviation of 1.2 mm at the coronal entry point, 1.4 mm at the apical end point, and an angular deviation of 3.5° [10]. Computer-aided surgery using dynamic navigation systems has shown lower mean horizontal deviations at the coronal entry point (0.71 ± 0.40 mm), apical end point (1.00 ± 0.49 mm), and angular deviation (2.26 ± 1.62°) [11]. Therefore, image-guided surgery approaches have been recommended to help increase the accuracy of ZI placement, preventing intraoperative and postoperative complications, as the length of ZIs is almost five times greater than a conventional-length dental implant; therefore, an entry-point or angular deviation of the dental implant bur may increase the apical-point deviation [12]. In addition, computer-aided surgery using dynamic navigation systems allows for free-hand implant navigation using high-precision motion tracking technology, preventing anatomical injuries [13]. Image-guided navigation systems also increase the accuracy of dental implant placement using artificial fiducial markers, which provide a virtual coordinate system linked to the surgical field or coordinate system of the patient [14]. However, computer-aided navigation systems are more expensive than surgical templates, and their accuracy depends on the learning curve and experience of the operator [15].

The objective of the present study was to evaluate the accuracy of placing ZIs using a dynamic navigation system. The null hypothesis (H0) stated that accuracy rates do not differ when comparing placement of ZIs using a dynamic navigation system versus a free-hand approach.

## 2. Materials and Methods

### 2.1. Study Design

This in vitro study was carried out between January and March 2021 at the Dental Center of Innovation and Advanced Specialties at Alfonso X El Sabio University in Madrid, Spain. The Ethical Committee of the Faculty of Health Sciences at Alfonso X El Sabio University approved the study in December 2020 (Process No. 25/2020). The patient gave their informed consent for the researchers to use their preoperative cone-beam computed tomography (CBCT) scan for the purposes of this study.

### 2.2. Experimental Procedure

Forty (40) ZIs (Galimplant, Sarria, Lugo, Spain) were planned and placed in teeth in positions 2.4 (4.3 mm × 30 mm, internal taper and conical wall), 2.2 (4.3 mm × 50 mm, internal taper and conical wall), 1.2 (4.3 mm × 52.5 mm, internal taper and conical wall), and 1.4 (4.3 mm × 35 mm, internal taper and conical wall). Researchers used an ANOVA to establish the sample size, achieving 80% power with a confidence level of 5%, with a variability between groups of 0.6 and an intragroup variability of 4, to identify differences in comparison with the null hypothesis H0: μ1 = μ2 = μ3 = μ4. Ten (10) anatomically based, standardized polyurethane models of a completely edentulous, atrophic, upper jaw maxilla were manufactured using a three-dimensional impression procedure (Sawbones Europe AB, Malmo, Sweden) based on a preoperative CBCT scan (WhiteFox, Satelec, Merignac, France). The scan was taken from a patient using the following exposure parameters: 8.0 mA, 105.0 kV peak, 7.20 s, with a 15 mm × 13 mm field of view (Figure 1). The anatomically based models were manufactured respecting the size and shape of the patient.

Afterwards, the models were fixed onto an artificial head to simulate the clinical conditions (Figure 2).

Researchers randomized the ZIs (Epidat 4.1, Galicia, Spain), which were assigned to one of two study groups: (A) ZI (Galimplant, Sarria, Lugo, Spain) placement using a computer-aided dynamic navigation system (Navident, ClaroNav, Toronto, ON, Canada) (*n* = 20) (navigation implant (NI)); and (B) manual ZI (Galimplant, Sarria, Lugo, Spain) placement using a free-hand technique (*n* = 20) (freehand implant (FHI)). The order of placement of the ZIs (Galimplant, Sarria, Lugo, Spain) was randomized across the study groups (Epidat 4.1, Galicia, Spain), beginning with the NI study group and followed by the FHI control group.

A preoperative CBCT scan was taken of the NI anatomically based standardized polyurethane models (WhiteFox, Satelec, Merignac, France) prior to placing a jaw tag; the use of polyurethane was based on the American Society for Testing and Materials’ (ASTM F-1839-08) approval of the use of polyurethane for testing instruments and dental implants (“Standard Specification for Rigid Polyurethane Foam for Use as a Standard Material for Test Orthopedic Devices for Instruments”) [16]. This black-and-white tag was affixed to the dental surface of the anatomically based, standardized polyurethane models using a photopolymerized composite resin (Navistent, ClaroNav, Toronto, ON, Canada). The datasets obtained from the CBCT scan were uploaded to treatment-planning software (Navident, ClaroNav, Toronto, ON, Canada) on a laptop computer mounted onto a mobile unit to simulate placement of the ZIs in accordance with the prior surgical planning (Figure 3A). An additional black-and-white drill tag was affixed to the handpiece (W & H, Bürmoos, Austria). The researchers calibrated and identified both of the optical reference markers with an optical triangulation tracking system using stereoscopic motion-tracking cameras, orienting the drilling process in real time to ensure that the planned angle, pathway, and depth were achieved. A ZI system (Galimplant, Sarria, Lugo, Spain) was used to perform the drilling, with this procedure being monitored using the computer-aided dynamic navigation system installed onto the laptop computer (Figure 3B).

The conventional free-hand technique was used to place all ZIs (Galimplant, Sarria, Lugo, Spain) that had been randomly assigned to the FHI control group, with the operator having access to the preoperative planning and CBCT scan. All ZIs (Galimplant, Sarria, Lugo, Spain) were placed by a unique operator with prior surgical experience.

### 2.3. Measurement Procedure

Following placement of the ZIs, the researchers conducted postoperative CBCT scans (WhiteFox, Satelec, Merignac, France) using the aforementioned exposure parameters. The planning and postoperative CBCT scans (WhiteFox, Satelec, Merignac, France) of the different study groups were subsequently imported into 3D implant-planning software (NemoScan, Nemotec, Madrid, Spain). The scans were then overlaid in order to assess the apical deviation, measured at the coronal entry point (mm), apical end point (mm), and angular deviation (°), with the latter measured at the center of the cylinder. Any deviations that were noted in any of the implants were subsequently analyzed and compared between the axial, sagittal, and coronal views (Figure 4A–C) by an independent operator. In addition, deviations in the positions of the zygomatic dental implants were also recorded and analyzed.

### 2.4. Statistical Analysis

For each of the response variables, tables were produced with summaries of the following statistics according to group, position and group, and position: number of observations, mean, standard deviation, median, and the minimum and maximum values. These were represented graphically by box plots. Linear regression models with repeated measures were adjusted to analyze the differences according to group, according to position, and the interaction between both variables. Where statistically significant differences were detected, two-to-two comparisons were made between groups/positions. The *p*-values were adjusted using the Tukey method to correct the type I error. The statistical analysis was carried out using the software SAS v.9.4 (SAS Institute Inc., Cary, NC, USA). Statistical decisions were made using a significance level of 0.05.

## 3. Results

Table 1 shows the mean, median, and SD values with 95% confidence intervals for the coronal entry point (mm), apical end point (mm), and angular deviations (°) of the NI study group and the FHI control group.

The paired Student’s *t*-test did not find any statistically significant differences in the coronal entry-point deviations between the study groups (*p* = 0.2617), nor in the ZI positions (*p* = 0.1649). However, statistically significant deviations were observed between the computer-aided dynamic navigation technique (NI) and the free-hand approach at the ZI position 2.4 (*p* = 0.0155) (Figure 5).

The paired Student’s *t*-test revealed statistically significant differences in the apical end-point deviations between the FHI control group and the NI study group (*p* = 0.0018). On the other hand, no statistically significant differences were observed between the zygomatic dental implant positions (*p* = 0.1856), except at the ZI position 2.4 (*p* = 0.0005) (Figure 6).

The paired Student’s *t*-test found no statistically significant differences between the angular deviations of the study groups (*p* = 0.3132); on the other hand, statistically significant differences were shown for the zygomatic dental implant positions (*p* = 0.0008), especially at the ZI position 1.4 (*p* = 0.0040) (Figure 7).

In summary, the FHI approach showed lower deviation values at the coronal entry point and the apical end point. This may be because the ZIs assigned to the FHI control group were the last to be placed, enabling the operator to learn and to memorize the correct position of the ZI. Furthermore, ZIs placed in posterior regions showed higher levels of deviation at the coronal entry point, apical end point, and angular level.

One ZI was withdrawn from the NI study group because the osteotomy site preparations did not provide sufficient stability for the ZI.

## 4. Discussion

The results of the present study reject the null hypothesis (H0), which states that there is no difference in accuracy when comparing placement of ZIs using a dynamic navigation system versus a free-hand approach.

The primary findings of the present study indicate that the free-hand conventional technique for ZI placement was more accurate than the computer-aided dynamic navigation technique at the coronal and apical levels; however, the computer-aided dynamic navigation technique showed more accuracy than the free-hand conventional technique at the angular level.

Hung et al. [17], Xiaojun et al. [18], Chen et al. [19], Hung et al. [20], Zhu et al. [8], and Jorba-García et al. [21] found computer-aided dynamic navigation to have greater accuracy than the free-hand conventional approach for the placement of conventional-length dental implants. Nevertheless, Brief et al. found that, although computer-aided navigation techniques are significantly more accurate than free-hand conventional techniques, the level of accuracy provided by the free-hand conventional technique is sufficient for most clinical situations [22]. However, Aydemir et al. reported that the conventional free-hand technique provides greater accuracy at the coronal entry-point and apical end-point levels than the computer-assisted dynamic navigation technique in the placement of ZIs, although the computer-aided dynamic navigation technique resulted in lower angular deviation than the free-hand conventional technique [23]. Moreover, Jung et al. found similar safety levels, outcomes, morbidity, and efficiency between computer-aided navigation techniques and free-hand conventional techniques for placement of conventional-length dental implants [24]. These results corroborate those of the present study, and the learning curve required for the use of computer-aided dynamic navigation systems may explain the discrepancies observed between the dynamic navigation system and the free-hand conventional technique [11]. Additionally, Mediavilla Guzmán et al. warned of low methodological quality in the studies related to ZIs, which makes it difficult to compare the results of the different studies [13]. Gunkel et al. also found that studies conducted under laboratory or in vitro conditions showed higher accuracy rates than clinical studies [25]. Likewise, Kim et al. [26] and Tahmaseb et al. [27] reported variability in the accuracy of computer-aided surgery using static navigation systems, depending on the study design.

Otherwise, Jorba-García et al. showed a mean angular deviation of 2.1° and a mean horizontal deviation of 0.46 mm at the coronal entry point for computer-aided surgery using dynamic navigation systems [28]. Xiaojun et al. showed a mean horizontal deviation of 1.36 ± 0.59 mm at the coronal entry point of conventional-length dental implants [18]; Chen et al., 1.12 ± 0.29 mm [14]; Hung et al., 1.07 ± 0.15 mm [20]; Hung et al., 1.35 ± 0.75 mm [17]; Block et al., 0.4 mm [29]; Kaewsiri et al., 1.05 ± 0.44 mm [30]; and Zhou et al., 1.56 ± 0.54 mm [8]. However, the present study showed a higher mean horizontal deviation at the coronal entry point (5.43 ± 2.13 mm), possibly due to the learning curve and operator experience. Kaewsiri et al. reported a mean horizontal deviation of 1.29 ± 0.50 mm at the apical end point, directly correlated with the length of the dental implant (8, 10, and 12 mm) [30]. Consequently, the horizontal deviation at the apical end point would be higher in ZIs than in conventional-length dental implants. In addition, Chrcanovic et al. reported an anteroposterior angular deviation of 8.06 ± 6.40° and craniocaudal of 11.20 ± 9.75°, which led to the invasion of the infratemporal fossa and the orbit by the ZIs [31]. Moreover, Vrielinck et al. found a mean angular deviation of 5.1° (ranging from 0.8 to 9.0° [32]; Xiaojun et al., 4.1 ± 0.9° [18]; Zhou et al., 2.52 ± 0.84° [8]; Hung et al., 1.37 ± 0.21° [17]; Hung et al., 2.05 ± 1.02° [20]; and Chen et al., 0.19 ± 0.92° [14]. These results are slightly lower than those obtained in the present study (7.36 ± 4.12°). These deviation values may lead to clinical and/or prosthetic complications in 36.4% of ZI placements, primarily due to a lack of primary stability [33]. In the present study, one ZI randomly placed using the dynamic navigation system was also removed due to insufficient stability at the prepared osteotomy site. In addition, Lan et al. described all complications related to the ZI placement procedure, reporting a malposition rate of 12% (95% confidence interval [CI]: 4 to 23%), surgical guiding failure rate of 11% (95% CI: 3 to 21%), and local infection/injury rate of 10% (95% CI: 3 to 18%) [34]. Additionally, Gutiérrez et al. reported that the prosthetic rehabilitations of ZIs have shown a low incidence of prosthetic complications (4.9% (95% CI: 2.7–7.3%)), regardless of the prosthetic treatment. The present study also found that the ZI placed in position 2.4 showed statistically significant deviations at the coronal entry point, and the ZI placed in position 1.4 showed statistically significant deviations in angular deviation. In summary, the ZIs placed in the anterior region showed lower deviations than the ZIs placed in the posterior region, likely due to better accessibility and visibility of the operative field.

The findings of the present study can be useful to clinicians in selecting the more accurate technique for ZI placement in patients with atrophied maxilla who must undergo full-arch rehabilitation by means of ZIs. The authors recommend improving the methodological quality of studies and increasing the body of evidence by way of additional randomized studies implementing new computer-assisted navigation techniques.

## 5. Conclusions

Within the limitations of the present study, the results indicated that the free-hand conventional technique provided greater accuracy in the placement of ZIs than the computer-assisted surgical technique. In addition, placement of ZIs in the anterior region was more accurate than that in the posterior region.

## Figures and Tables

**Figure 1 jcm-11-01436-f001:**
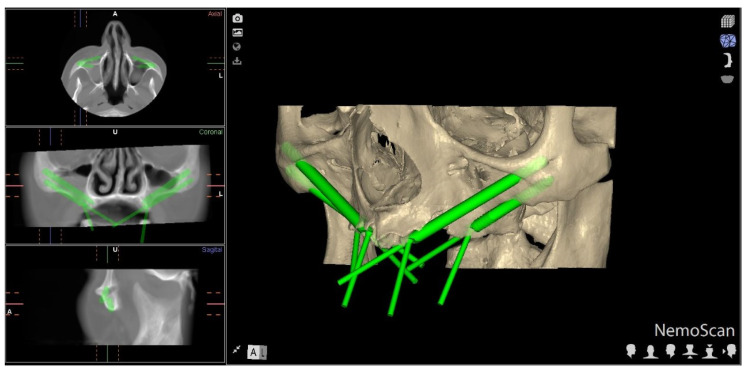
Preoperative planning of ZI placement based on the CBCT scan taken of the patient. Detail of the coronal, sagittal, and axial views, and three-dimensional reconstruction.

**Figure 2 jcm-11-01436-f002:**
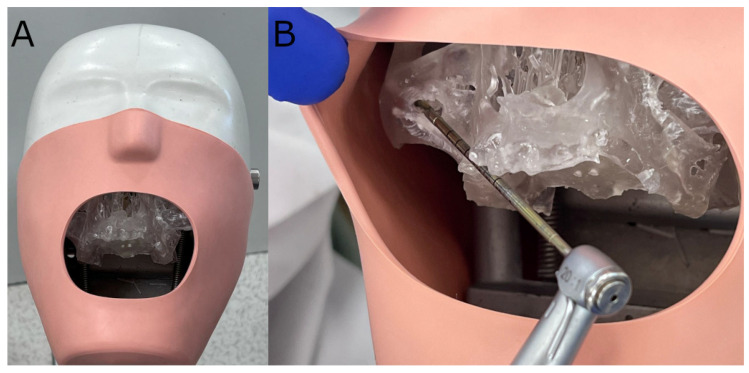
(**A**) Polyurethane model fixed onto an artificial head and (**B**) drilling procedure.

**Figure 3 jcm-11-01436-f003:**
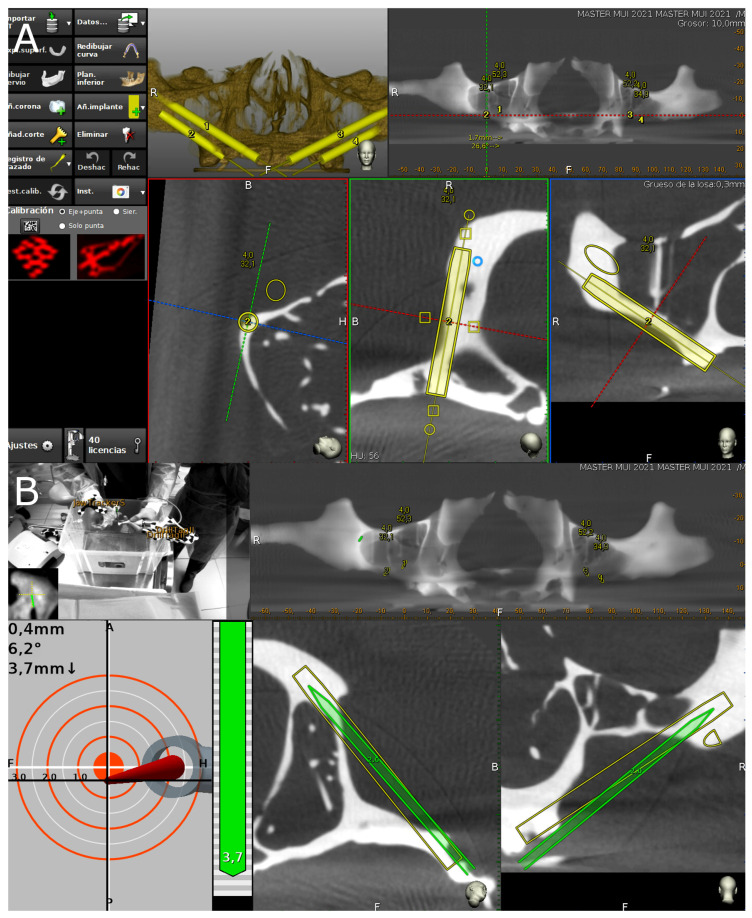
(**A**) Preoperative planning of placement of ZIs with the dynamic navigation appliance using treatment-planning software, and (**B**) tracking procedure during ZI placement.

**Figure 4 jcm-11-01436-f004:**
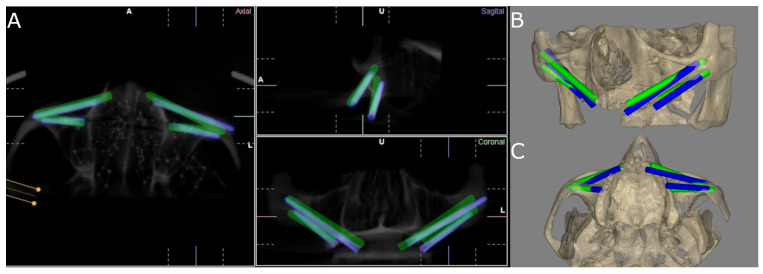
(**A**) CBCT images of the coronal, sagittal, and axial views; and (**B**) front and (**C**) bottom views of the three-dimensional reconstruction of the measurement procedure contrasted against the preoperative planning (green cylinders) and postoperative ZI placement (blue cylinders) of the ZIs placed on the experimental model.

**Figure 5 jcm-11-01436-f005:**
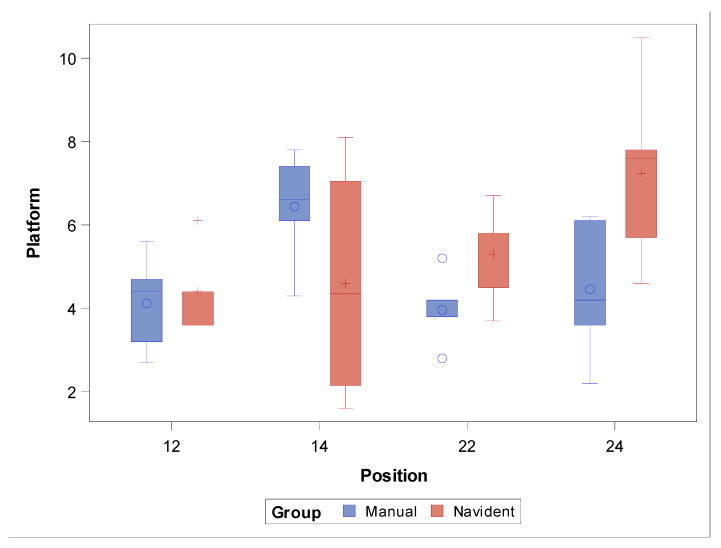
Box plot of deviations at the coronal entry point observed in the study groups and ZI positions. Median values are represented by the horizontal lines within each box. The symbols represent extreme values.

**Figure 6 jcm-11-01436-f006:**
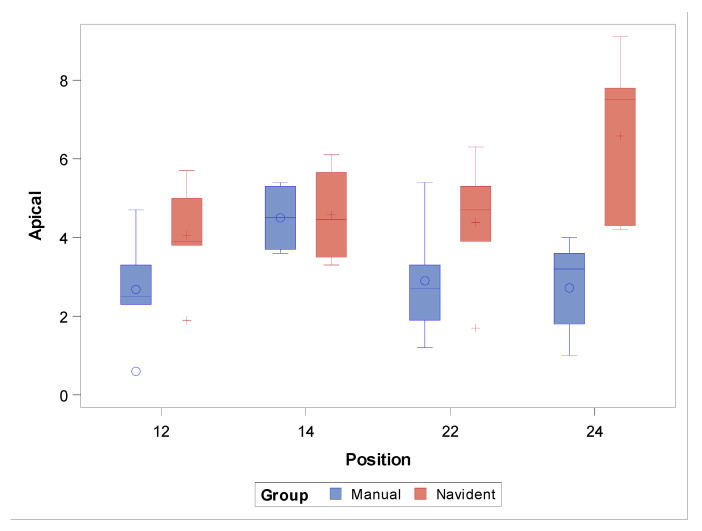
Box plot of apical end-point deviations recorded in the study groups and ZI positions. Median values are represented by the horizontal lines within each box. The symbols represent extreme values.

**Figure 7 jcm-11-01436-f007:**
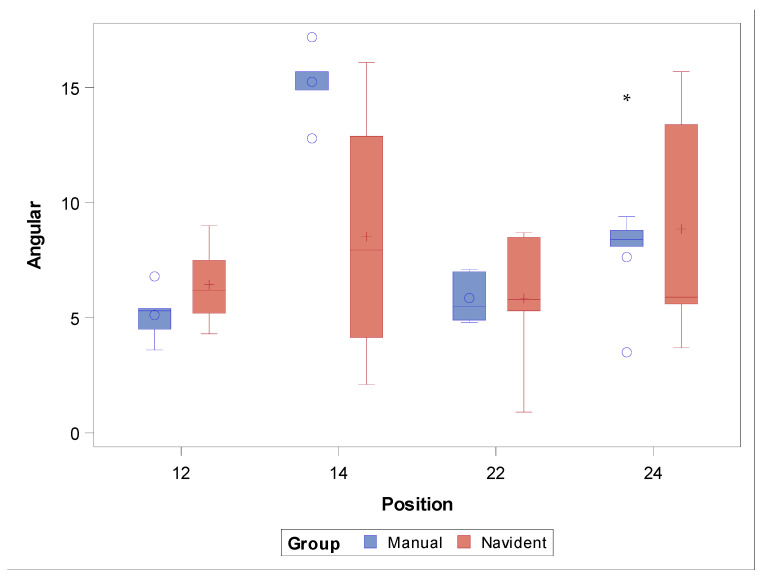
Box plot of angular deviations found in the study groups and ZI positions. Median values are represented by the horizontal lines within each box. The symbols represent extreme values.

**Table 1 jcm-11-01436-t001:** Descriptive values of deviations at the coronal entry point (mm), apical end point (mm), and angular (°) deviations of the computer-aided dynamic navigation technique (NI) and the free-hand approach.

		*n*	Mean	Median	SD	Lower 95% CL for Mean	Upper 95% CL for Mean	Minimum	Maximum
CORONAL	NI	19	5.43 ^a^	5.70	2.13	4.41	6.46	1.60	10.50
	FHI	20	4.75 ^a^	4.35	1.58	4.01	5.48	2.20	7.80
APICAL	NI	19	4.92 ^a^	4.70	1.89	4.00	5.83	1.70	9.10
	FHI	20	3.20 ^b^	3.30	1.45	2.52	3.88	0.60	5.40
ANGULAR	NI	19	7.36 ^a^	6.20	4.12	5.37	9.34	0.90	16.10
	FHI	20	8.47 ^b^	7.05	4.40	6.41	10.53	3.50	17.20

^a,b^ Statistically significant differences (*p* < 0.05) found between groups. NI: navigation implants; and FHI: free-hand implants.

## Data Availability

Data are available on request, pursuant to any applicable restrictions (e.g., ethical or privacy considerations).

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
