# Peer review of "Accuracy of a Computer-Aided Dynamic Navigation System in the Placement of Zygomatic Dental Implants: An In Vitro Study"

_jcm, 2022, doi:10.3390/jcm11051436_

Round 1

Reviewer 1 Report

The study carried out by the authors has practical value for modern implantology.

However, there are a few points that I would like to clear.

  1. The accuracy of many dental manipulations, including surgical operations, strictly depends on the ergonomics of their behavior. Obviously, at the real “live” operation on a patient, the accuracy of both “free-hand” surgery and dynamic navigation surgery will depend on the patient’s head position, macro-and micro-movements of the head and upper body.  How and where did you install/fix the model of the edentulous jaw during the in-vitro operation? How did you simulate the conditions of the “real” surgical operations? This is a very important point since it influences the results.
  2. Why this type of composite material was chosen for the manufacture of the jaw model? What type of bone tissue does it resemble? Several studies have shown a correlation between the accuracy of the surgical operation and the type of bone tissue. Would you please explain?
  3. Why did you not compare the dynamic navigation system, the free-hand method, and the placement of implants using surgical guides?

Author Response

Dear Reviewer 1,

I’m pleased to resubmit the manuscript of the work entitled, “Accuracy of Computer-Aided Dynamic Navigation System in Placement of Zygomatic Dental Implants. An In Vitro Study”

Reviewer 1: English language and style are fine/minor spell check required

Response: In order to adapt to the reviewer's 1 comments, we have sent the manuscript to the English Editing Service of MDPI. We attached the Certificate.

Reviewer 1: The accuracy of many dental manipulations, including surgical operations, strictly depends on the ergonomics of their behavior. Obviously, at the real “live” operation on a patient, the accuracy of both “free-hand” surgery and dynamic navigation surgery will depend on the patient’s head position, macro-and micro-movements of the head and upper body.  How and where did you install/fix the model of the edentulous jaw during the in-vitro operation? How did you simulate the conditions of the “real” surgical operations? This is a very important point since it influences the results.

Response: In order to adapt to the reviewer's 1 comments, we have added a sentence and a Figure in the Material and Methods section, specifying that: “the models were fixed on an artificial head trying to simulate the clinical conditions”.

Reviewer 1: Why this type of composite material was chosen for the manufacture of the jaw model? What type of bone tissue does it resemble? Several studies have shown a correlation between the accuracy of the surgical operation and the type of bone tissue. Would you please explain?

Response: In order to adapt to the reviewer's 1 comments, we have added a sentence and reference in the Material and Methods section to justify the selection of polyurethane material: “the American Society for Testing and Materials (ASTM F-1839-08) approved the use of polyurethane for testing instruments and dental implants ("Standard Specification for Rigid Polyurethane Foam for Use as a Standard Material for Test Orthopedic Devices for Instruments”) (Comuzzi L, Tumedei M, Pontes AE, Piattelli A, Iezzi G. Primary Stability of Dental Implants in Low-Density (10 and 20 pcf) Polyurethane Foam Blocks: Conical vs Cylindrical Implants. Int J Environ Res Public Health. 2020, 17, 2617. doi: 10.3390/ijerph17082617.)”.

Reviewer 1: Why did you not compare the dynamic navigation system, the free-hand method, and the placement of implants using surgical guides?

Response: In order to adapt to the reviewer's 1 comments, we appreciate the suggestion made by the reviewer and clarify that we will carry out this study in the future

We take this opportunity to thank the recommendations and suggestions made by the reviewers to improve the document.

Yours sincerely,

Reviewer 2 Report

This is an in vitro study investigating the accuracy of zygomatic implant placement via free-hand and dynamic-guidance techniques.

The study has major study flaws which must be repeated in the guidance of a biostatistician and experienced scientist.

Major concerns: There is no study category as described by “randomized controlled experimental trial “ also “International Organization for Standardization (ISO 14801). “ is a quality control organisations which has no relevance with this study. Please revise or remove these sentences.

If this is a clinical trial (as described in the manuscript) please revise accordingly If this is a study done on polyurethane models (described in the abstract and M&M)  (an in vitro investigation) please remove the sentences pertinent to clinical trial.

If it was been correctly understood, the clinicians scanned the skull of a patient via CBCT and printed a polyurethane model out of this DICOM data. Then planned ZI on these model. It is unclear how was the shape and size of this model as well as the setting of the experiment which authors used to place implants

Authors mention “Drilling order was also randomized (Epidat 4.1, Galicia, Spain), beginning with the NI study group and followed by the FHI control group. “. Was this another randomisation regarding commercial drills? If so what was the hypothesis to randomise the order of drills? It does not make sense to start tih the final diameter for the osteotomy of a zygomatic implant.

The statistical analysis conducted are not appropriate for this study since the deviations of the implants placed in the same model by the same operator is not independent and errors cluster on the same model and/or operator. Please consult to a biostatistician for the proper statistical analysis of dependent data.

Please take a photograph of the experimental setting with the operator and the model.

Author Response

Dear Reviewer, 2

I’m pleased to resubmit the manuscript of the work entitled, “Accuracy of Computer-Aided Dynamic Navigation System in Placement of Zygomatic Dental Implants. An In Vitro Study”

Reviewer 2: Extensive editing of English language and style required

Response: In order to adapt to the reviewer's 1 comments, we have sent the manuscript to the English Editing Service of MDPI. We attached the Certificate.

Reviewer 2: There is no study category as described by “randomized controlled experimental trial

Response: In order to adapt to the reviewer's 2 comments, we have changed the description of the study design.

Reviewer 2: “International Organization for Standardization (ISO 14801). “ is a quality control organisations which has no relevance with this study.

Response: In order to adapt to the reviewer's 2 comments, we have removed the sentence.

Reviewer 2: If this is a clinical trial (as described in the manuscript) please revise accordingly If this is a study done on polyurethane models (described in the abstract and M&M)  (an in vitro investigation) please remove the sentences pertinent to clinical trial.

Response: In order to adapt to the reviewer's 2 comments, we have clarified that it is an in vitro study. We have changed the description of the study design in the manuscript.

Reviewer 2: If it was been correctly understood, the clinicians scanned the skull of a patient via CBCT and printed a polyurethane model out of this DICOM data. Then planned ZI on these model. It is unclear how was the shape and size of this model as well as the setting of the experiment which authors used to place implants

Response: In order to adapt to the reviewer's 2 comments, we have added two sentences in the Material and Method section.

Reviewer 2: Authors mention “Drilling order was also randomized (Epidat 4.1, Galicia, Spain), beginning with the NI study group and followed by the FHI control group. “. Was this another randomisation regarding commercial drills? If so what was the hypothesis to randomise the order of drills? It does not make sense to start tih the final diameter for the osteotomy of a zygomatic implant.

Response: In order to adapt to the reviewer's 2 comments, we clarify that the randomization was carried out to decide the order of the study groups, since the learning acquired by the operator during the placement of the implants can influence the precision of the last implants. We have made a change in the manuscript.

Reviewer 2: The statistical analysis conducted are not appropriate for this study since the deviations of the implants placed in the same model by the same operator is not independent and errors cluster on the same model and/or operator. Please consult to a biostatistician for the proper statistical analysis of dependent data.

Response: In order to adapt to the reviewer's 2 comments, we clarify that the analysis was performed using a model for repeated measures. These models take into account that one or more measurements are taken on the same subject. This analysis was performed with SAS Proc Mixed.

Reviewer 2: Please take a photograph of the experimental setting with the operator and the model.

Response: In order to adapt to the reviewer's 2 comments, we have added a photograph (Figure 2).

We take this opportunity to thank the recommendations and suggestions made by the reviewers to improve the document.

Yours sincerely,

Round 2

Reviewer 2 Report

The changes have substantially improved the quality and clarity of the manuscript. The design of the study related to the skull scan of one single patient imposes further speculation but these results merit publication as a pilot study. 

The tables should be reformatted for uniformity. It would be important to add 95% confidence intervals to the results.

Author Response

Dear Reviewer 2,

I’m pleased to resubmit the manuscript of the work entitled, “Accuracy of Computer-Aided Dynamic Navigation System in Placement of Zygomatic Dental Implants. An In Vitro Study”

Reviewer 2: The tables should be reformatted for uniformity. It would be important to add 95% confidence intervals to the results.

Response: In order to adapt to the reviewer's 2 comments, we clarify that there is only one Table (Table 1). We have added the 95% confidence intervals to the Table 1 of the Results section.

We take this opportunity to thank the recommendations and suggestions made by the reviewers to improve the document.

Yours sincerely,